# UniMem: Towards a Unified View of Long-Context Large Language Models

**Junjie Fang**[4]*, **Likai Tang**[1]*, **Hongzhe Bi**[1], **Yujia Qin**[1], **Si Sun**[1], **Zhenyu Li**[1], **Haolun Li**[1], **Yongjian Li**[1], **Xin Cong**[1], **Yankai Lin**[5], **Yukun Yan**[1], **Xiaodong Shi**[4†], **Sen Song**[1], **Zhiyuan Liu**[1,2,3†], **Maosong Sun**[1,2,3]

[1]Department of Computer Science and Technology, Institute for AI, Tsinghua University, China
[2]Beijing National Research Center for Information Science and Technology, China
[3]Institute for AI Industry Research (AIR), Tsinghua University, Beijing, China
[4]Xiamen University, [5]Renmin University of China
`fangjj@stu.xmu.edu.cn, tanglk20@mails.tsinghua.edu.cn`

## Abstract

Long-context processing is a critical ability that constrains the applicability of large language models (LLMs). Although there exist various methods devoted to enhancing the long-context processing ability of LLMs, they are developed in an isolated manner and lack systematic analysis and integration of their strengths, hindering further developments. In this paper, we introduce **UniMem**, a **Uni**fied framework that reformulates existing long-context methods from the view of **Mem**ory augmentation of LLMs. Distinguished by its four core dimensions—Memory Management, Memory Writing, Memory Reading, and Memory Injection, **UniMem** empowers researchers to conduct systematic exploration of long-context methods. We re-formulate 16 existing methods based on UniMem and analyze four representative methods: Transformer-XL, Memorizing Transformer, RMT, and Longformer into equivalent UniMem forms to reveal their design principles and strengths. Based on these analyses, we propose **UniMix**, an innovative approach that integrates the strengths of these algorithms. Experimental results show that UniMix achieves superior performance in handling long contexts with significantly lower perplexity than baselines. The code is publicly available at https://github.com/thunlp/UniMem

## 1 Introduction

Transformer-based (Vaswani et al., 2017) large language models (LLMs) have ushered in a new era of AI (Brown et al., 2020; Touvron et al., 2023; Bommasani et al., 2021; OpenAI, 2022), leading to various applications (e.g., natural language processing (Brown et al., 2020), code generation (Li et al., 2022)). As the scope of LLM applications expands, the demand for handling longer contexts becomes paramount (e.g., parsing long documents (Gao et al., 2023) and managing intricate dialogues (Zheng et al., 2023)). However, the inherent computational limitations of traditional Transformer architectures, stemming from the quadratic complexity of self-attention mechanisms, pose significant challenges for scaling to long contexts.

Various methods have been developed to address this challenge, including optimizing the quadratic complexity of Transformer Dai et al. (2019); Beltagy et al. (2020); Wu et al. (2022), adjusting the position encoding of Transformer to handle longer contexts than during pre-training (Chen et al., 2023a; Peng et al., 2023), and designing non-Transformer architectures (Gu et al., 2021; Xiong et al., 2023; Gu & Dao, 2023; Sun et al., 2023). Among them, the first type of method has been extensively explored and utilized, but lacks integration among its various types, prompting our primary focus on it. Upon reviewing this type of

---

*   Indicates equal contribution.
†   Corresponding authors: `mandel@xmu.edu.cn, liuzy@tsinghua.edu.cn`.

| UniMem Framework | | | | | |
|---|---|---|---|---|---|
| Method | Memory Management | | Memory Writing | Memory Reading | Memory Injection |
| | Memory Size | Overflow Handling | | | |
| RMT Bulatov et al. (2022) | Single-Sgm | FIFO | Model forward | All | All-Lyr |
| AutoCompressor Chevalier et al. (2023) | Multi-Sgm | / | Model forward | All | All-Lyr |
| Poolingformer Zhang et al. (2021) | Multi-Sgm | Clear All | Pooling+Direct | Position | All-Lyr |
| LongT5 Guo et al. (2021) | Multi-Sgm | Clear All | Pooling+Direct | Position | All-Lyr |
| Longformer Beltagy et al. (2020) | Multi-Sgm | Clear All | Direct | Position | All-Lyr |
| Big Bird Zaheer et al. (2020) | Multi-Sgm | Clear All | Direct | Position | All-Lyr |
| LongNet Ding et al. (2023) | Multi-Sgm | Clear All | Direct | Position | All-Lyr |
| Transformer-XL Dai et al. (2019) | Single-Sgm | FIFO | Direct | Position | All-Lyr |
| StreamingLLM Xiao et al. (2023) | Multi-Sgm | FIFO | Direct | Position | All-Lyr |
| Routing Roy et al. (2021) | Multi-Sgm | Clear All | Direct | Similarity | All-Lyr |
| Reformer Kitaev et al. (2020) | Multi-Sgm | Clear All | Direct | Similarity | All-Lyr |
| Memorizing Transformer Wu et al. (2022) | Multi-Sgm | FIFO | Direct | Similarity | Certain-Lyr |
| FoT Tworkowski et al. (2023) | Multi-Sgm | FIFO | Direct | Similarity | Certain-Lyr |
| Unilimiformer Bertsch et al. (2023) | Multi-Sgm | / | Direct | Similarity | All-Lyr |
| KNN-LM Khandelwal et al. (2019) | Multi-Sgm | / | Direct | Similarity | Certain-Lyr |
| TRIME Zhong et al. (2022) | Multi-Sgm | / | Direct | Similarity | Certain-Lyr |
| UniMix (Ours) | Multi-Sgm | FIFO | Direct+Model forward | Similarity+Position | Certain-Lyr |

Table 1: Long-context methods are categorized under four dimensions using our **UniMem** framework (Section 3), introducing **UniMix** to combine strengths. Excluding FoT, KNN-LM, and TRIME—focused on inference—other methods apply to both training and inference.

method, we further divide it into three categories: (1) *Context Caching*, which stores the intermediate hidden states of context and retrieves relevant ones when processing successive contexts (Wu et al., 2022; Dai et al., 2019; Bertsch et al., 2023; Tworkowski et al., 2023); (2) *Context Compression*, which compresses the context into condensed tokens and prepends them before the successive context (Bulatov et al., 2022; Chevalier et al., 2023; Guo et al., 2021; Zhang et al., 2021); (3) *Sparse Attention*: which elaborately designs sparse attention masks to reduce computational complexity to extend input length (Beltagy et al., 2020; Zaheer et al., 2020; Ding et al., 2023; Kitaev et al., 2020; Roy et al., 2021; Chen et al., 2023b).

While the above approaches effectively reduce computational complexity and enhance long-context capabilities, they still face the following problems. Firstly, there is a notable **absence of equitable evaluation** across various approaches. Given that these approaches adhere to distinct design principles and are tested within disparate datasets and settings, assessing their efficacy and adaptability fairly becomes exceedingly complex. Such complexity complicates the comparative analysis to identify the strengths of each approach. Secondly, the diverse implementations of these approaches significantly **impede the strength integration** of each one into a singular model. This impediment to synergistic integration restricts the potential for substantial progress in the effective processing of long contexts by LLMs.

This work introduces **UniMem**, a unified framework that integrates existing long-context methods for optimizing Transformer computation complexity, viewed via memory augmentation. Specifically, UniMem consists of four essential dimensions: (1) *Memory Management* determines how much past information is stored and how old memory is replaced, impacting how LLM recalls and uses previous context; (2) *Memory Writing* describes how the model converts recent information into a memory format, affecting the way that past data is summarized and accessed; (3) *Memory Reading* focuses on how the model retrieves information from the memory bank, crucial for efficiently using stored information to understand the current context; (4) *Memory Injection* determines which model layers to augment memory information, influencing overall efficiency and effectiveness in processing long contexts.

Based on the UniMem framework, as illustrated in Table 1, we re-formulate 16 prevailing long-context approaches from the unified view of the four fundamental dimensions. We further select 4 representative approaches from each long-context category: Transformer-XL (Dai et al., 2019), Memorizing Transformer (Wu et al., 2022), RMT (Bulatov et al., 2022), and Longformer (Beltagy et al., 2020), and introduce their equivalent formula under the UniMem framework. This unification facilitates a clearer understanding of their subtle interconnections and distinct characteristics. Our analysis indicates a shared foundation in design principles across these methodologies, despite their superficial differences. Additionally, built upon the unified framework, we introduce a new method **UniMix** which synthesizes the strengths of three categories of long-context algorithms. The experimental results demonstrate that UniMix can achieve superior performance compared to the existing long-context approaches, including position interpolation techniques. Furthermore, our investigation into UniMix reveals two beneficial conclusions: (1) By integrating different memory dimensions, UniMix demonstrates strong robustness and is less sensitive to hyper-

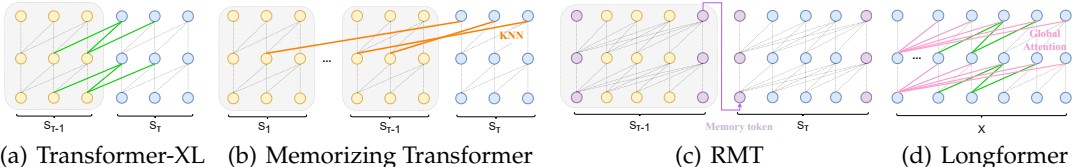

(a) Transformer-XL   (b) Memorizing Transformer        (c) RMT              (d) Longformer

Figure 1: Diagram illustrates long-context methods (segment length $L = 3$). Yellow circles show past segments; blue circles mark the current segment. (a) Transformer-XL caches earlier hidden states. (b) Memorizing Transformer retrieves past segments with kNN similarity. (c) RMT employs memory tokens for prior segments. (d) Longformer extends segments with global and sliding window attention.

parameters. (2) Increasing the number of memory layers does not necessarily correlate with performance gains. Rather, the strategic positioning of a single memory layer can achieve comparable effectiveness to integrating additional memory across half of the model's layers.

## 2 Preliminaries

### 2.1 Vanilla Transformer

Given a Transformer-based LLM featuring $N$ stacked layers, and an input sequence $X = \{x_1, x_2, ..., x_T\}$ consisting of $T$ tokens, the LLM processes the sequence $X$ layer-wisely, constructing a set of hidden states $\mathbf{H}^n = \{\mathbf{h}_1^n, \mathbf{h}_2^n, ..., \mathbf{h}_T^n\}$, where $n \in [1, N]$. The $n$-th LLM layer's hidden states $\mathbf{H}^n$ are derived from the $(n-1)$-th layer using feed-forward networks (FFN), multi-head attention (Attn), and softmax (Sftx):

$$\mathbf{H}^n \overset{\text{FFN}}{\longleftarrow} \text{Attn}(\mathbf{Q}^n, \mathbf{K}^n, \mathbf{V}^n; \mathbf{A}) = \text{Sftx}\left(\frac{\mathbf{Q}^n(\mathbf{K}^n)^T}{\sqrt{d}} + \mathbf{A}\right)\mathbf{V}^a, \ \mathbf{A}_{ij} = \begin{cases} 0 \text{ if } j \in [0, i], \\ -\infty \text{ else.} \end{cases} \quad (1)$$

$$\mathbf{Q}^n / \mathbf{K}^n / \mathbf{V}^n = \mathbf{H}^{n-1}(\mathbf{W}_q^n)^T / \mathbf{H}^{n-1}(\mathbf{W}_k^n)^T / \mathbf{H}^{n-1}(\mathbf{W}_v^n)^T. \quad (2)$$

where $\mathbf{A} \in \mathbb{R}^{T \times T}$ is often a causal mask matrix controlling the visible context field and $\mathbf{W}_{q/k/v}^n \in \mathbb{R}^{d \times d}$ is the projection matrix of multi-head attention in $n$-th LLM layer.

Despite LLMs' strong context modeling, their attention mechanism's $T^2$ computational complexity challenges tasks with very long contexts. To alleviate this challenge, the community has made many improvements, and we analyze three main types of long-context approaches including context caching, context compression, and sparse attention methods.

### 2.2 Context Caching

Context caching methods innovatively tackle long-context modeling by splitting them into segments, and storing the hidden states of historical segments and retrieves relevant ones when processing the current segment. To be specific, a long input sequence $X = \{x_1, x_2, ..., x_T\}$ can be split into a succession of segments $S_\tau = \{x_{(\tau-1) \times L+1}, x_{(\tau-1) \times L+2}, ..., x_{(\tau-1) \times L+L}\}$ with the fixed token length $L$ ($L \ll T$), where $\tau \in [1, T/L]$. These methods utilize not only the hidden states from the current segment but also those from previous segments during sequential processing. Broadly, there are two categories: one involves storing the hidden states of a previous segment in an external memory cache (Dai et al., 2019), and the other involves caching key-value pairs of multiple previous segments (Wu et al., 2022; Bertsch et al., 2023; Tworkowski et al., 2023). This paper highlights Transformer-XL (Dai et al., 2019) and Memorizing Transformers (Wu et al., 2022) as exemplars to demonstrate how external memory integration enhances the capacity for long-context modeling for these two categories.

**Transformer-XL** utilizes both hidden states $\mathbf{H}_\tau^n$ of the current segment $S_\tau$ and the hidden states $\mathbf{H}_{\tau-1}^n$ processed by the previous segment $S_{\tau-1}$, as shown in Figure 1(a). This recurrence way enables Transformer-XL to learn dependencies beyond long contexts of fixed

length without breaking temporal coherence. Formally, the $n$-th hidden states $\mathbf{H}_\tau^n$ of segment $S_\tau$ is constructed layer-wisely as follows:

$$\mathbf{H}_\tau^n \overset{\text{FFN}}{\longleftarrow} \text{Attn}(\mathbf{Q}_\tau^n, \widetilde{\mathbf{K}}_\tau^n, \widetilde{\mathbf{V}}_\tau^n; \mathbf{A}), \ \mathbf{A}_{ij} = \begin{cases} 0 \text{ if } j \in (i, i+L), \\ -\infty \text{ else.} \end{cases} \tag{3}$$

$$\widetilde{\mathbf{K}}_\tau^n / \widetilde{\mathbf{V}}_\tau^n = [\text{SG}(\mathbf{H}_{\tau-1}^{n-1}); \mathbf{H}_\tau^{n-1}](\mathbf{W}_k^n)^T / [\text{SG}(\mathbf{H}_{\tau-1}^{n-1}); \mathbf{H}_\tau^{n-1}](\mathbf{W}_v^n)^T, \tag{4}$$

where mask matrix $\mathbf{A} \in \mathcal{R}^{L \times 2L}$ is illustrated in Figure 2 (a). The function SG$(\cdot)$ means to stop gradient back-propagation and $[\cdot; \cdot]$ denotes the concatenation operation[1].

**Memorizing Transformer** designates a specific layer for memory augmentation. When modeling each segment $S_\tau$, the modeling process for non-memory extension layers aligns with the vanilla Transformer (Eq. 1). In the memory augmentation layer, as depicted in Figure 1(b), each query token $\mathbf{q}_i^n \in \mathbf{Q}_\tau^n$ can access additional $k$ pairs of memory keys and values from all historical processed keys $\mathbf{K}_{1:\tau\text{-}1}^n = \{\mathbf{k}_1^n, \mathbf{k}_2^n, ..., \mathbf{k}_{(\tau-1)\times L}^n\}$ and values $\mathbf{V}_{1:\tau\text{-}1}^n = \{\mathbf{v}_1^n, \mathbf{v}_2^n, ..., \mathbf{v}_{(\tau-1)\times L}^n\}$ through $k$-nearest-neighbor ($k$NN) lookup[2]:

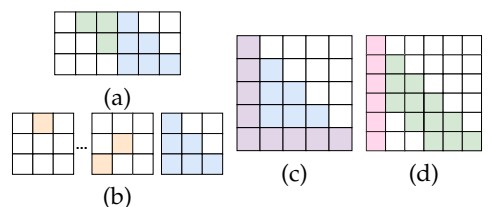

Figure 2: Attention patterns for long-context methods ($L = 3$).(a) Transformer-XL. (b) Memorizing Transformer. (c) RMT. (d) Longformer.

$$\mathbf{H}_\tau^n \overset{\text{FFN}}{\longleftarrow} \mathbf{g} \odot \text{Attn}(\mathbf{Q}_\tau^n, \text{SG}(\mathbf{K}_{1:\tau\text{-}1}^n), \text{SG}(\mathbf{V}_{1:\tau\text{-}1}^n); \hat{\mathbf{A}}) + (1 - \mathbf{g}) \odot \text{Attn}(\mathbf{Q}_\tau^n, \mathbf{K}_\tau^n, \mathbf{V}_\tau^n; \mathbf{A}), \ \hat{\mathbf{A}}_{ij} = \begin{cases} 0 \text{ if } j \in k\text{NN}(i), \\ -\infty \text{ else.} \end{cases} \tag{5}$$

where $\mathbf{g}$ is a learnable parameter and $\odot$ denotes element-wise multiplication. The mask matrix (Figure 2 (b)) $\hat{\mathbf{A}} \in \mathcal{R}^{L \times ((\tau-1) \times L)}$, $j \in k\text{NN}(i)$ means the dot product of the $j$-th previous key of $\mathbf{K}_{1:\tau\text{-}1}^n$ and the $i$-th current query token of $\mathbf{Q}_\tau^n$ ranks in the top $k$ positions among the dot product of the $i$-th current query token and all previous key tokens. The second mask $\mathbf{A} \in \mathcal{R}^{L \times L}$ is the vanilla causal mask.

## 2.3 Context Compression

Context compression methods typically compress the hidden states of history tokens into high-level memory representations, which combine multiple tokens into one concise representation. This technique can be implemented through either a pooling-based approach, exemplified by LongT5 (Guo et al., 2021) and PoolingFormer (Zhang et al., 2021), or a model forward strategy, such as Recurrent Memory Transformer (RMT) (Bulatov et al., 2022) and AutoCompressor(Chevalier et al., 2023). We choose RMT to demonstrate how compressed memory representations are integrated into the modeling process.

**RMT** incorporates $m$ tunable memory tokens "[mem]" before and after each segment $S_\tau$ (Figure 1(c)). The succeeding memory tokens (marked as $\mathbf{M}_\tau^n$) are employed as a medium to read the historical information from the last segment $S_{\tau\text{-}1}$ while the succeeding memory tokens ($\overline{\mathbf{M}}_\tau^n$) can be used to write the current segment information for modeling next segment $S_{\tau+1}$. In this way, the $n$-th hidden states of segment $S_\tau$ is expanded to $\widetilde{\mathbf{H}}_\tau^n = [\mathbf{M}_\tau^n; \mathbf{H}_\tau^n; \overline{\mathbf{M}}_\tau^n]$:

$$\widetilde{\mathbf{H}}_\tau^n \overset{\text{FFN}}{\longleftarrow} \text{Attn}(\widetilde{\mathbf{Q}}_\tau^n, \widetilde{\mathbf{K}}_\tau^n, \widetilde{\mathbf{V}}_\tau^n; \mathbf{A}), \ \mathbf{A}_{ij} = \begin{cases} 0 \text{ if } j \in [0, \max(i, m)], \\ 0 \text{ elif } i \in [L+m, L+2m], \\ -\infty \text{ else.} \end{cases} \tag{6}$$

Where the mask matrix $\mathbf{A} \in \mathcal{R}^{(L+2m) \times (L+2m)}$ is illustrated in Figure 2 (c). To implement the recurrence mechanism, RMT initializes the first layer representation of read and write memory tokens $\mathbf{M}_\tau^1$ and $\overline{\mathbf{M}}_\tau^1$ of current segment $S_\tau$ with the last layer representations of write memory tokens $\overline{\mathbf{M}}_{\tau\text{-}1}^N$ of last segment $S_{\tau\text{-}1}$.

---

[1]$\mathbf{H}_\tau^{n-1}$ is spliced after $\mathbf{H}_{\tau-1}^{n-1}$, resulting in $\mathbf{A}_{ij} = 0$ if $j \in (i, i+L]$.

[2]If the memory pool size ($m$) is limited, only the $\tau - m$ to $\tau - 1$ historical segments can be accessed.

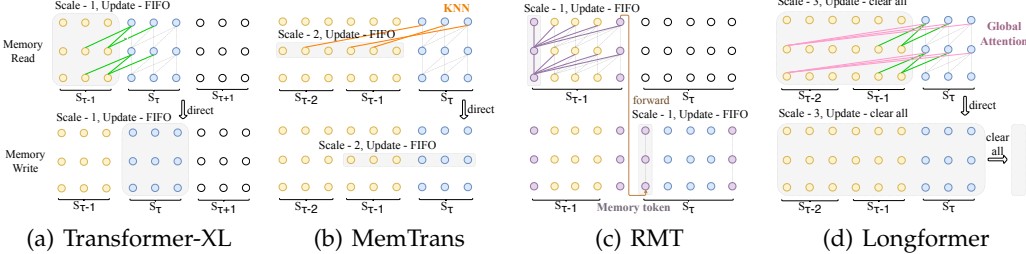

Figure 3: Four typical methods in UniMem. (a) Transformer-XL uses a single past hidden state, shown as memory cache, with sliding window attention. (b) Memorizing Transformers (MemTrans) saves multiple segment states in a layer, using kNN similarity for retrieval. (c) RMT utilizes a memory token for prior segment data, stored post-model forwarding. (d) Longformer applies global and sliding window attention based on token positioning.

## 2.4 Sparse Attention

Besides segment-level methods, sparse attention techniques tackle long-context tasks by sparse mechanisms to avoid the quadratic computational increase of vanilla attention with long sequences Beltagy et al. (2020); Zaheer et al. (2020); Ding et al. (2023); Kitaev et al. (2020); Roy et al. (2021). Formally, given an input sequence $X = \{x_1, x_2, ..., x_L\}$, these methods modify the standard attention mechanism by creating diverse sparse attention masks:

$$\mathbf{H}^n \xleftarrow{\text{FFN}} \text{Attn}(\mathbf{Q}^n, \mathbf{K}^n, \mathbf{V}^n; \mathbf{A}), \qquad (7)$$

where $\mathbf{H}^n \in \mathcal{R}^{T \times d}$ is the $n$-th layer hidden states of $X$ and $\mathbf{A} \in \mathcal{R}^{T \times T}$ is the mask matrix. We illustrate sparse attention mask construction using Longformer (Beltagy et al., 2020) as an example. See Appendix A.1 for BigBird details.

**Longformer** utilizes a combination of sliding window and global attention to establish the sparse attention pattern (Figure 1(d)). Sliding window attention constrains attention for each token within a context of $L$ tokens (We set window length to be equal to segment length and ignore stride here), while global attention allows any token in the sequence to capture global tokens $G$. Hence, its mask matrix (Figure 2 (d)) can be formulated as:

$$\mathbf{A}_{ij} = \begin{cases} 0 \text{ if } j \in [\min(0, i\text{-}L), i] \cup \text{index}(G), \\ -\infty \text{ else.} \end{cases} \qquad (8)$$

# 3 The Unified Memory Framework

Through reviewing the aforementioned methods, we investigate that these methods can be integrated into a unified memory-augmented modeling framework, featuring a segment-level streaming input mode. We name this framework as **UniMem**. In this section, we first reformulate the aforementioned long-context modeling methods within the UniMem framework, distinguishing them across precisely defined memory dimensions, and then integrate multiple memory dimensions to propose new modeling approaches.

## 3.1 UniMem Formulation

Formally, UniMem can augment the memory keys $\mathbf{K}^n_{\text{mem}}$ and values $\mathbf{V}^n_{\text{mem}}$ from previously processed segments when modeling the hidden state $\mathbf{H}^n_\tau$ for each segment $S_\tau$:

$$\mathbf{H}^n_\tau \xleftarrow{\text{FFN}} \text{Attn}(\mathbf{Q}^n_\tau, [\mathbf{K}^n_{\text{mem}}; \mathbf{K}^n_\tau], [\mathbf{V}^n_{\text{mem}}; \mathbf{V}^n_\tau]; \mathbf{A}). \qquad (9)$$

The distinctions among existing long-context modeling methods hinge on two critical elements: the construction of memory keys and values ($\mathbf{K}^n_{\text{mem}}/\mathbf{V}^n_{\text{mem}}$), and the creation of the attention mask matrix to access memory ($\mathbf{A}$). Next, we present the aforementioned long-context modeling methods in UniMem, focusing on constructing these essential elements.

**UniMem: Transformer-XL**, as shown in Figure 3(a), designates the keys and values from the last segment $S_{\tau\text{-}1}$ as memory keys and values $\mathbf{K}^n_{\text{mem}}/\mathbf{V}^n_{\text{mem}} = \text{SG}(\mathbf{K}^n_{\tau\text{-}1})/\text{SG}(\mathbf{V}^n_{\tau\text{-}1})$, with

the mask remaining consistent with that in Eq. 3. Transformer-XL considers the keys $\mathbf{K}_{\tau\text{-}1}^n$ and values $\mathbf{V}_{\tau\text{-}1}^n$ from the last segment $S_{\tau\text{-}1}$ as the memory keys and values:

$$\mathbf{K}_{\text{mem}}^n / \mathbf{V}_{\text{mem}}^n = \text{SG}(\mathbf{K}_{\tau\text{-}1}^n) / \text{SG}(\mathbf{V}_{\tau\text{-}1}^n), \ \mathbf{A}_{ij} = \begin{cases} 1 \text{ if } j \in (i, i+L], \\ -\infty \text{ else.} \end{cases} \tag{10}$$

**UniMem: Memorizing Transformer**, as illustrated in Figure 3(b), preserves the key-value pairs of all processed segments as memory key and values $\mathbf{K}_{\text{mem}}^n / \mathbf{V}_{\text{mem}}^n = \text{SG}(\mathbf{K}_{1:\tau\text{-}1}^n) / \text{SG}(\mathbf{V}_{1:\tau\text{-}1}^n)$ and accesses the memory using $k$NN's attention mechanism:

$$\mathbf{A}_{ij} = \begin{cases} 0 \text{ if } j \in k\text{NN}(i) \cup [(\tau\text{-}1) \times L, (\tau\text{-}1) \times L + i], \\ -\infty \text{ else.} \end{cases} \tag{11}$$

Given the UniMem formulation, we can deduce the original formulation of Memorizing Transformer in reverse (we omit attention masks for simplicity):

$$\begin{aligned} \text{Attn}(\mathbf{Q}_\tau^n, [\mathbf{K}_{\text{mem}}^n; \mathbf{K}_\tau^n], [\mathbf{V}_{\text{mem}}^n; \mathbf{V}_\tau^n]) &= \text{Sftx}\left(\mathbf{Q}_\tau^n[\mathbf{K}_{\text{mem}}^n; \mathbf{K}_\tau^n]^T\right)[\mathbf{V}_{\text{mem}}^n; \mathbf{V}_\tau^n], \\ &= \mathbf{g} \odot \text{Sftx}\left(\mathbf{Q}_\tau^n(\mathbf{K}_{\text{mem}}^n)^T\right)\mathbf{V}_{\text{mem}}^n + (1 - \mathbf{g}) \odot \text{Sftx}\left(\mathbf{Q}_\tau^n(\mathbf{K}_\tau^n)^T\right)\mathbf{V}_\tau^n, \\ &= \mathbf{g} \odot \text{Attn}(\mathbf{Q}_\tau^n, \mathbf{K}_{\text{mem}}^n, \mathbf{V}_{\text{mem}}^n) + (1 - \mathbf{g}) \odot \text{Attn}(\mathbf{Q}_\tau^n, \mathbf{K}_\tau^n, \mathbf{V}_\tau^n), \end{aligned} \tag{12}$$

where $\mathbf{g}$ is constructed as follows:

$$\mathbf{g} = \frac{\sum_{k=1}^{(\tau-1) \times L} \exp(\mathbf{Q}_\tau^n(\mathbf{K}_{\text{mem}}^n)^T)_k}{\sum_{k=1}^{(\tau-1) \times L} \exp(\mathbf{Q}_\tau^n(\mathbf{K}_{\text{mem}}^n)^T)_k + \sum_{l=1}^{L} \exp(\mathbf{Q}_\tau^n(\mathbf{K}_\tau^n)^T)_l}. \tag{13}$$

There is also experiment evidence that two implementations can be converted to each other without loss of language modeling performance Tworkowski et al. (2023).

**UniMem: RMT** constructs the memory keys and values $\mathbf{K}_{\text{mem}}^n / \mathbf{V}_{\text{mem}}^n$ by performing a model forward pass on the read memory token $\mathbf{M}_\tau^n$ within the current segment $S_\tau$:

$$\mathbf{Q}_{\text{mem}}^n / \mathbf{K}_{\text{mem}}^n / \mathbf{V}_{\text{mem}}^n = \mathbf{M}_\tau^{n-1}(\mathbf{W}_q^n)^T / \mathbf{M}_\tau^{n-1}(\mathbf{W}_k^n)^T / \mathbf{M}_\tau^{n-1}(\mathbf{W}_v^n)^T, \ \mathbf{M}_\tau^{n-1} \xleftarrow{\text{FFN}} \text{Attn}(\mathbf{Q}_{\text{mem}}^{n-1}, \mathbf{K}_{\text{mem}}^{n-1}, \mathbf{V}_{\text{mem}}^{n-1}), \tag{14}$$

where $\mathbf{Q}_{\text{mem}}^{n-1}$ denotes memory token queries for retrieving key-value pairs, $\mathbf{K}_{\text{mem}}^n / \mathbf{V}_{\text{mem}}^n$, in the model forward pass. The entire modeling process is shown in Figure 3(c).

**UniMem: Longformer** uses sliding window attention, like other sparse attention methods, to handle long sequences $X = x_1, x_2, ..., x_T$, where each token attends to a limited number of previous tokens. This approach facilitates transitioning input into a streaming format, processing fixed-length segments $S_1, S_2, ..., S_\tau$ sequentially. Longformer also applies global attention based on sliding window attention, as illustrated in Figure 3(d). The memory keys and values $\mathbf{K}_{\text{mem}}^n / \mathbf{V}_{\text{mem}}^n$ are derived from global tokens $G$ and the last segment $S_\tau$:

$$\mathbf{K}_{\text{mem}}^n / \mathbf{V}_{\text{mem}}^n = [\mathbf{K}_{\text{glob}}^n; \mathbf{K}_{\tau\text{-}1}^n] / [\mathbf{V}_{\text{glob}}^n; \mathbf{V}_{\tau\text{-}1}^n], \ \mathbf{A}_{ij} = \begin{cases} 0 \text{ if } j \in (i, i+L+|G|], \\ -\infty \text{ else,} \end{cases} \tag{15}$$

where $|G|$ is the number of global tokens and $\mathbf{K}_{\text{glob}}^n / \mathbf{V}_{\text{glob}}^n$ are their key-value pairs.

### 3.2 Memory Dimensions

In UniMem, the differences among these methods can be further standardized into four dimensions related to operating the added memory keys and values (Table 1).

**Memory Management**: This dimension manages memory cache storage, involving two crucial elements: *(1) Memory Size* refers to the amount of stored memory key-value pairs from previous segments. Some methods store from just one prior segment ("Single-Sgm"), such as RMT and Transformer-XL as Figure 3(a) and 3(c) illustrates, whereas other methods implement a large memory cache incorporating multiple past segments ("Multi-Sgm"), (e.g. Memorizing Transformer and Sparse Attention, as depicted in Figure 3(b) and 3(d)). *(2) Overflow Handling* denotes the strategies for updating the memory cache when it is full. Some use First-In-First-Out ("FIFO") to discard old memory, while sparse attention methods clear all when max capacity is reached (Figure 3(d)).

**Memory Writing**: This dimension concerns transforming processed data from past segments into memory keys and values for storage and selective access by the current segment.

| Model | Memory Management | Memory Reading | | | Memory Writing | Mem Injection |
|-------|-------------------|------|---------------|---------------|----------------|---------------|
| | Memory Size | Topk | Window Length | Global Tokens | Compressed Tokens | Memory Layer |
| **Vanilla** | 0 | 0 | 0 | 0 | 0 | - |
| **Longformer** | 4,096 | 0 | 2,048 | 4 | 0 | All |
| **MemTrans** | 20,480 | 32 | 0 | 0 | 0 | 11,21 |
| **Trans-XL** | 2,048 | 0 | 2,048 | 0 | 0 | All |
| **RMT** | 40 | 0 | 0 | 0 | 40 | All |
| **UniMix** | 20,520 | 4 | 2,048 | 4 | 40 | 12-22 |

Table 2: Long-context methods decomposed along the defined UniMem dimensions in hyperparameters view for TinyLLaMA (Section 3.2).

Context Caching and Sparse Attention insert keys and values directly during the forward pass ("Direct"), as shown in Figure 3(b). Conversely, Context Compression employs operations like dual forward passes (e.g., RMT, "Model Forward") and pooling (e.g., Poolingformer (Zhang et al., 2021), LongT5 (Guo et al., 2021), "Pooling") to compress the last segment's data into tokens, dictated by the *Compressed Tokens* count.

**Memory Reading**: This dimension explains fetching keys and values from the memory cache, guided by mask matrix $A$ (Eq. 9). The "Position" method (e.g., Longformer) selects based on relative positions, influenced by *Window Length* and *Global Tokens* (Eq.15). The "Similarity" method (e.g., Memorizing Transformer, Routing Transformer) uses similarity to current queries, regulated by *Topk* (Eq. 11). RMT accesses the whole cache ("All"). Some models (e.g., KNN-LM (Khandelwal et al., 2019), TRIME (Zhong et al., 2022)) also map memory cache contents to vocabulary space, integrating them with current tokens.

**Memory Injection**: This aspect defines how LLM layers add an external memory cache, controlled by the *Memory Layer* parameter for its quantity and placement. While most methods integrate memory uniformly across all layers ("All-Lyr"), selectively adding memory to specific layers can lower CUDA memory usage ("Certain-Lyr").

**Others**: Besides design elements, methods vary in training strategies. Transformer-XL and Memorizing Transformer use stop-gradient to block memory gradients, while RMT and Longformer enable back-propagation through time (BPTT). Additionally, Focused Transformer uses cross-batch training, and TRIME utilizes in-batch contrastive loss.

## 3.3 UniMix: Synthesizing Different Dimensions

We analyzed existing methods from the UniMem perspective, as shown in Figure 3. From this analysis, we developed **UniMix**, which merges the best features in each Memory Dimension. UniMix utilizes "Similarity" and "Position" for memory reading and a "Direct" plus "Model Forward" approach for memory writing to yield empirically robust—but not necessarily optimal—results. This approach serves as a starting point for further exploration. Ongoing research will assess UniMix's performance, the effects of different dimensions on outcomes, and the potential of this new dimensional synergy for enhanced results.

# 4 Experiments

## 4.1 Experiments Settings

**Datasets and Evaluation** We use the language modeling task as the testbed. We select two widely-used datasets: (1) PG-19 (Rae et al., 2019), from the Project Gutenberg archive, consisting of pre-1919 English books (Sun et al., 2021); and (2) GitHub dataset, including a broad collection of code and documentation from RedPajama's GitHub repositories (Computer, 2023). We apply perplexity as the metric for evaluation.

**Implementations** We evaluate two scales of LLaMA: the 22-layer TinyLLaMA-1.1B (Zhang et al., 2024) and 32-layer LLaMA2-7B (Touvron et al., 2023). Unless specifically stated, our experiments predominantly utilized the TinyLLaMA model. We fine-tune models for one epoch using 0.1B tokens across two datasets (We have found that this amount of training volume is sufficient for achieving solid long-context capabilities and stable method

comparisons). The default sequence length was $2,048$ tokens, aligned with TinyLLaMA's positional embeddings. See Appendix D for more implementation details.

**Baselines** We compare our UniMix with vanilla Transformer (Vaswani et al., 2017), Longformer (Beltagy et al., 2020), Memorizing Transformers (MemTrans) (Wu et al., 2022), Transformer-XL (Dai et al., 2019), RMT (Bulatov et al., 2022), and positional interpolation (Chen et al., 2023a). We re-implement all of the existing methods and our UniMix under the UniMem framework. See Table 2 for detailed hyperparameters.

## 4.2 Comparison of Existing Methods Under UniMem

The results are shown in Table 3, from which we conclude that: (1) We compare the existing classical methods under the same setting and find that the performance trend remains consistent across different datasets. The performance ranking is Transformer-XL > Longformer > MemTrans > RMT. We hope this comparative result can serve as a reference for the community. (2) Our UniMix method outperforms or matches existing methods in both text and code datasets, demonstrating that by effectively organizing each dimension and combining optimization of various methods, we can achieve a better long-context processing method that is applicable across various fields. (3) The UniMix method remains superior on the larger-scale LLaMA2-7B model, emphasizing the method's scalability and robustness.

| Model | TinyLLaMA | | LLaMA2-7B | |
|---|---|---|---|---|
| | PG19 | Github | PG19 | Github |
| **Vanilla** | 14.53 | 2.66 | 11.01 | 2.53 |
| **Longformer** | 13.92 | 2.42 | 10.34 | 2.27 |
| **MemTrans** | 14.34 | 2.57 | 10.66 | 2.34 |
| **Transformer-XL** | **13.78** | 2.36 | **10.28** | 2.23 |
| **RMT** | 14.68 | 2.70 | 10.64 | 2.55 |
| **UniMix** | **13.78** | **2.32** | **10.28** | **2.20** |

Table 3: Perplexity of different methods in UniMem.

## 4.3 Effects of Different UniMem Dimensions

The UniMem framework integrates various approaches to long-context language modeling, each characterized by distinct hyperparameters. In this section, our objective is to dissect the impact of each dimension within the UniMem framework. Subsequent analyses have focused on discerning the pivotal role of Memory Read and Memory Injection within the **UniMix** method by fine-tuning associated hyperparameters. Experiments on Memory Management and Memory Writing adjustments are detailed in the Appendix B.

**Memory Reading** Our study delved into the hyperparameters affecting memory reading—specifically, *Topk* and *Window Length*—and their impact on model perplexity (PPL). Key insights are listed as follows: *(1) TopK Increment Does Not Guarantee Improved Model Performance*: Increasing TopK from 0 to 64 did not uniformly boost performance. While the Memorizing Transformer's PPL steadily dropped with higher TopK, in the UniMix framework (MemTrans settings), optimal performance was noted at TopK=4 and 8, with negligible benefits beyond these points, as illustrated in Figure 4. *(2) Window Length's Minimal Impact on UniMix*: Adjusting the window length from 0 to 2048 had little effect on UniMix, unlike Transformer-XL, which showed significant performance changes. This highlights a distinct contrast in how these models respond to window length adjustments. *(3) UniMix's Robustness*: The observations underscore UniMix's resilience against single-dimensional parameter shifts like TopK and window length, marking a notable strength for long-context processing. Such robustness minimizes the necessity for intricate hyperparameter optimization, ensuring stable performance across diverse settings.

**Memory Injection** To investigate the impact of Memory Injection dimensions on performance, we conducted further analysis on the GitHub dataset. Our findings are summarized as follows: *(1) Considerable Impact of Memory Layer Placement*: For the TinyLLaMA model, inserting the memory layer at higher levels outperforms its insertion at lower levels, as demonstrated in Figure 4(c). *(2) Significant Influence of Single Memory Layer Position*: The positioning of the memory layer has a substantial effect on performance, with an optimal layer significantly outperforming others, as shown in Figure 4(d). *(3) Task-Independent Optimal*

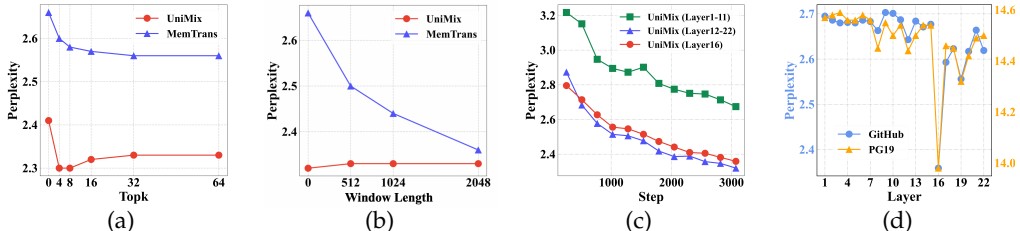

Figure 4: Effects of different UniMem dimensions on perplexity across datasets. (a) *Topk*'s role for MemTrans and UniMix; (b) Combined effects with *Window Length*; (c) Memory Layer Distribution's impact; (d) Memory Layer Position's influence (Single Layer Injection).

*Memory Layer*: The optimal layer remains consistent across different datasets (GitHub and PG19) for the same model configuration, according to Figure 4(d). *Model-Dependent Optimal Memory Layer*: On the GitHub dataset, the optimal layer for the 7b model is the 11th layer, whereas, for the 1b model, it's the 16th layer. Notably, the 1b model has a singular optimal layer, whereas multiple optimal layers exist for the 7b model, as illustrated in Figure 4(d) and Figure 10. *(4) The superiority of Optimal Memory Layer in Practical Applications*: When inserting a single memory layer across different levels of the model, the performance at the 16th layer significantly surpasses that of other levels. Remarkably, the enhancement achieved by a single layer at the 16th level is comparable to the improvements observed when memory layers are inserted across layers 12 to 22, far exceeding the performance improvements seen when inserted in layers 1 to 11. This indicates that a single memory layer, optimally positioned, can reach more than 98% of full performance using just 1/11 the computational time and resources (Figure 4(d)).

## 4.4 Comparison with Position Interpolation

We evaluate methodologies for processing longer texts, comparing their performance on a GitHub dataset with segments over 64k (Table 4). Our experiment involved: 1) Extending vanilla transformers to 64k inputs and fine-tuning; 2) Using Position Interpolation (PI) for increased input to 64k by adjusting position indices; 3) Implementing UniMix, which processes up to 64k, combining 62k memory and 2k local attention. After fine-tuning on a 0.1B token dataset, UniMix showed significant perplexity

| Model | Perplexity |
|---|---|
| Vanilla-64K | 20.57 |
| Position Interpolation | 5.87 |
| UniMix | **2.75** |

Table 4: Perplexity comparison among Vanilla, Position Interpolation (PI), and UniMix models.

reduction over PI, benefiting from a memory strategy similar to the LLaMA model's pretraining, without new positional embeddings. Crucially, UniMix demonstrates a computational complexity that scales linearly with text length, thereby surpassing the Vanilla Transformer and PI in terms of inference and training efficiency due to their exponential time complexity.

## 4.5 Downstream Task Experiments

Downstream task experiments were conducted on the TinyLLaMA model with a context length of 512 tokens and a UniMix memory size of 10k tokens. We proportionally adjusted other hyperparameters based on those specified in the original paper. The process begins with fine-tuning the model on a 0.1B token dataset. Subsequently, we apply supervised fine-tuning (SFT) using 12k instances from the LongAlpaca dataset. Although both fine-tuning and SFT experiments are moderate in scope, potentially affecting the models' performance on downstream tasks, the results still demonstrate the expected performance trends.

**Results on LongBench** As shown in Table 5, The results of the downstream tasks indicate that the performance ranking almost aligns with the perplexity (PPL) evaluation in our paper: UniMix >Transformer-XL >Longformer >RMT >MemTrans. This consistency across different evaluation metrics highlights the robustness of our proposed UniMix method.

| Model | NQA | Qasper | MFQA | HQA | 2WikiMQA | Musique |
|---|---|---|---|---|---|---|
| Vanilla | 5.2 | 8.1 | 19.45 | 6.19 | **16.6** | 3.58 |
| Longformer | 4.5 | 8.01 | **19.52** | 6.11 | 15.74 | **3.8** |
| MemTrans | 5.88 | **8.69** | 19.18 | 5.55 | 14.12 | 3.49 |
| Transformer-XL | 4.54 | 8.11 | 18.98 | 5.15 | 14.33 | 3.23 |
| RMT | 4.36 | 7.58 | 18.59 | 5.9 | 13.75 | 3.78 |
| **UniMix (Ours)** | **6.19** | 8.26 | 19.27 | **6.82** | 14.45 | 3.77 |

| Model | GovReport | QMSum | MultiNews | TREC | TQA | SAMSum |
|---|---|---|---|---|---|---|
| Vanilla | 12.47 | 19 | 10.06 | 11 | 11.9 | 0.42 |
| Longformer | 9.68 | 19.45 | 11.28 | 16.5 | 14.54 | 0.17 |
| MemTrans | 10.84 | 18.72 | 13.14 | 12 | 11.82 | 0.0 |
| Transformer-XL | 13.33 | 19.6 | **13.57** | 13 | **17.79** | 0.0 |
| RMT | 11.6 | 19.54 | 11.08 | 13 | 12.33 | 0.0 |
| **UniMix (Ours)** | **16** | **20.02** | 12.09 | **20** | 12.24 | **3.01** |

| Model | PsgCount | PsgRetrieval | LCC | RB-P | Avg |
|---|---|---|---|---|---|
| Vanilla | 0.0 | **4.0** | 25.34 | 32.53 | 11.62 |
| Longformer | 1.0 | **4.0** | 24.56 | 31.55 | 11.90 |
| MemTrans | **2.75** | 0.25 | 26.48 | 30.78 | 11.48 |
| Transformer-XL | 1.12 | 0.92 | 24.75 | 32.38 | 11.93 |
| RMT | 1.55 | 1.0 | 28.4 | 32.02 | 11.53 |
| **UniMix (Ours)** | 1.62 | 2.5 | **28.37** | **33.65** | **13.01** |

Table 5: Performance of Different Methods on LongBench.

| Model | 4k | 8k | 16k | 32k | Avg |
|---|---|---|---|---|---|
| Vanilla | 0.05 | 0.04 | 0.03 | 0 | 0.03 |
| Longformer | 0.07 | 0.03 | 0.03 | 0 | 0.03 |
| MemTrans | 0.05 | 0 | 0 | 0 | 0.01 |
| Transformer-XL | 0.20 | 0.04 | 0.03 | 0.01 | 0.07 |
| RMT | 0.04 | 0.03 | 0.03 | 0.01 | 0.03 |
| **UniMix (Ours)** | **0.37** | **0.39** | **0.09** | **0.04** | **0.22** |

Table 6: Accuracy of different methods on Needles in the Haystack Dataset.

**Needles in the Haystack Dataset** In addition to our evaluations on LongBench, we conduct experiments using the "Needles in the Haystack" dataset. This dataset includes text lengths ranging from 4k to 32k tokens, as shown in Table 6. UniMix is the only method to accurately identify the "needles" in texts exceeding 4k tokens in length.

# 5 Conclusion

In this paper, we introduce UniMem to unify various long-context approaches under the view of memory augmentation of LLMs. UniMem encompasses four dimensions: memory management, memory writing, memory reading, and memory injection. UniMem provides a structured categorization that enhances understanding, revealing, for instance, that StreamingLLM and Memorizing Transformer differ mainly in their memory reading techniques. This framework allows researchers to optimize long-context modeling by exploring different strategies like Least Recently Used (LRU) for Memory Management and various compression formats for Memory Writing. Our findings show that the memory injection layer significantly impacts performance, with each language model having an optimal layer independent of the dataset. By enabling granular control through tunable hyperparameters, UniMem ensures standardized and fair evaluations. Furthermore, we propose UniMix, which combines the strengths of these methods and significantly outperforms existing approaches across diverse datasets. Future work will integrate UniMem with other techniques, such as diverse position encoding and alternative overflow handling methods, focusing on differential memory strategies across layers.

## Acknowledgement

This work is supported by the National Key R&D Program of China (No.2022ZD0116312), National Natural Science Foundation of China (No. 62236004), the Postdoctoral Fellowship Program of CPSE (Grant No. GZB20230343), China Postdoctoral Science Foundation (Grant No. 2023M741945), National Science and Technology Major Project (Grant No. 2022ZD0116101), the Key Support Project of NSFC-Liaoning Joint Foundation (Grant No. U1908216), the Major Scientific Research Project of the State Language Commission in the 13th Five-Year Plan (Grant No. WT135-38) and the public technology service platform project of Xiamen City (Grant No. 3502Z20231043). Yujia Qin is sponsored by the Baidu Scholarship.

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

# A  Existing Method

## A.1  BigBird

**Big Bird** Zaheer et al. (2020) introduces random attention in addition to sliding window attention and global token attention, i.e., it randomly selects some key tokens for each query token where the query can attend over these keys:

$$\mathbf{A}_{ij} = \begin{cases} 0 \text{ if } j \in [\min(0, i\text{-}L), i] \cup G \cup R(i), \\ -\infty \text{ else.} \end{cases} \quad (16)$$

where $R(i)$ denotes the position set of randomly selected key tokens for the $i$-th query token.

**UniMem: Big Bird** can also be reformulated as the following memory-augmented modeling process:

$$\mathbf{K}^n_{\text{mem}} / \mathbf{V}^n_{\text{mem}} = [\mathbf{K}^n_{\text{glob}}; \mathbf{K}^n_{\text{rand}}; \mathbf{K}^n_{\tau\text{-}1}] / [\mathbf{V}^n_{\text{glob}}; \mathbf{V}^n_{\text{rand}}; \mathbf{V}^n_{\tau\text{-}1}],$$

$$\mathbf{A}_{ij} = \begin{cases} 0 \text{ if } j \in [i, i + L + |G| + |R|], \\ -\infty \text{ else,} \end{cases} \quad (17)$$

where $|R|$ is the number of random attention tokens and $\mathbf{K}^n_{\text{rand}} / \mathbf{V}^n_{\text{rand}}$ are their keys and values.

# B  Effects of Different UniMem Dimensions

## B.1  Memory Management

We focus on Overflow Handling in the Memory Management dimension. Specifically, we tune the Overflow Handling dimension from default First-In-First-Out ("FIFO") to "Clear all" for Longformer, Memtrans, and Mix. We find that Longformer and UniMix achieve worse perplexity with "Clear all", which can be the result of the reduced horizon of attention of "Clear all" (Figure 5). On the other hand, Memtrans gets slightly better perplexity with "Clear all". We hypothesize that this could have originated from the "Memory staleness problem" as in Wu et al. (2022).

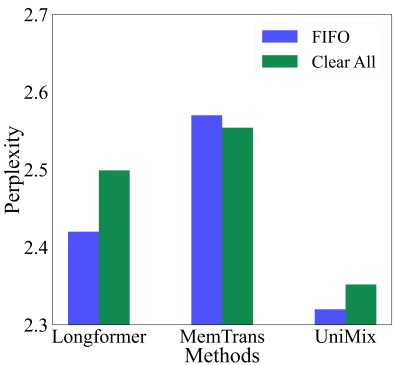

Figure 5: Impact of *Overflow Handling* on perplexity for Longformer, MemTrans and UniMix.

## B.2  Memory Write

We adjust the "Memory Tokens" for UniMix. to alter the way Memory writes are conducted. The larger the "Memory Tokens", the more memory is written in the form of Model forward. When the "Memory Tokens" is set to 0, all memory writes are direct. It can be observed in Figure 6 that increasing the "Memory Tokens" does not demonstrate a positive effect. This is consistent with the experimental results of existing methods discussed earlier.

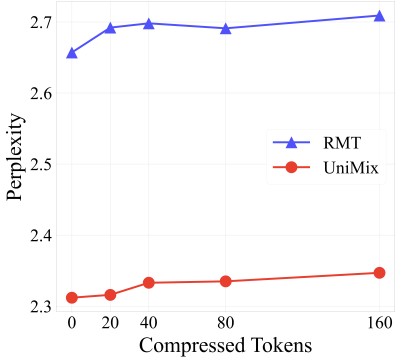

Figure 6: Impact of *Compressed Tokens* on perplexity for UniMix.

Figure 7: Perplexity of Layer(1-7), Layer(8-14) and Layer(15-21) on UniMix

## B.3 Memory Injection

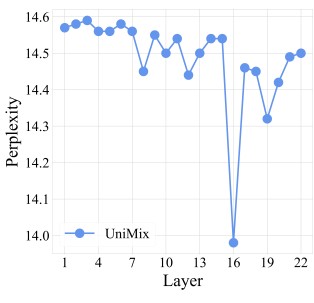
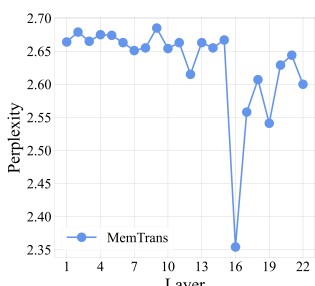
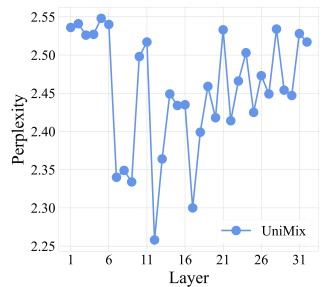

Figure 8: Perplexity of Single Layer on UniMix(PG19, TinyLLaMA)

Figure 9: Perplexity of Single Layer on Memtrans(Github, TinyLLaMA)

Figure 10: Perplexity of Single Layer on UniMix(Github, LLaMA2-7B).

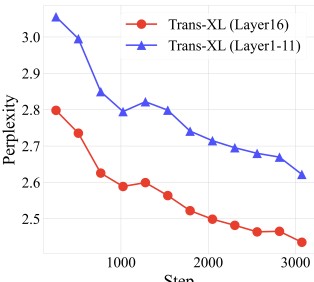
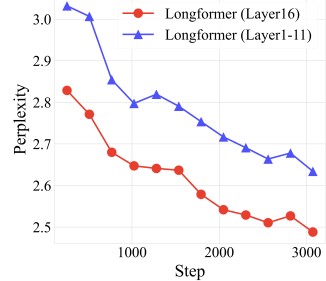
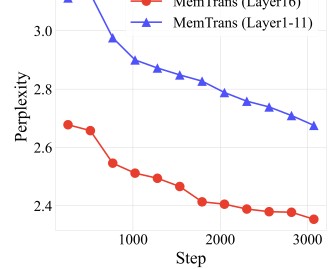

Figure 11: Perplexity of Layer(1-11) and Layer(16) on Transformer-XL (Github).

Figure 12: Perplexity of Layer(1-11) and Layer(16) on Longformer (Github).

Figure 13: Perplexity of Layer(1-11) and Layer(16) on Memtrans (Github).

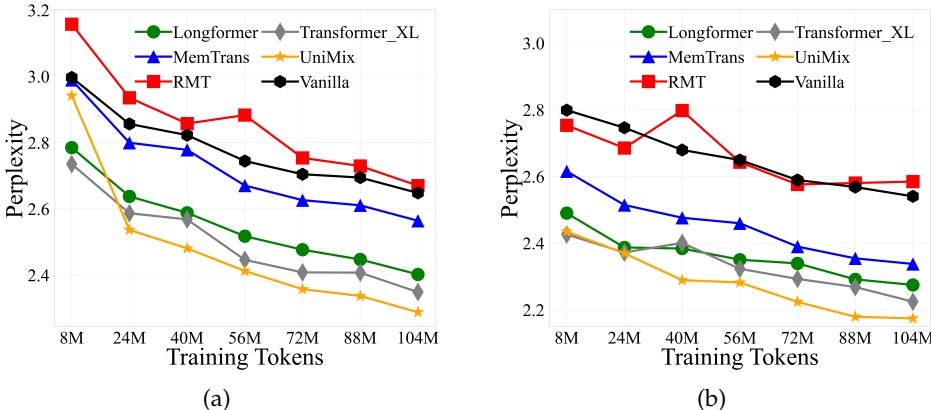

Figure 14: Perplexity Curves of Different Methods with Varying Training Token Amounts on the GitHub Dataset. (a) TinyLLaMA model. (b) LLaMA2-7B model.

## C  Impact of different training data scales

We explored the impact of varying fine-tuning data volumes on long-context modeling methods. We plotted the Perplexity (PPL) for TinyLLAMA and LLAMA2-7B using different scales of training data, as shown in Figure 14. Our results indicate that increasing the volume of training tokens significantly reduces perplexity and enhances model performance. UniMix consistently emerged as the most effective method across all data scales. Moreover, methods such as UniMix and Transformer-XL demonstrated excellent scalability, suggesting their potential for even better performance with larger datasets.

## D  Hyperparameters

| Hyperparameter | Value | |
|---|---|---|
| | **Common** | |
| Learning rate | 5E-5 | |
| Learning rate schedule | Linear | |
| Optimizer | AdamW | |
| DeepSpeed Config | ZeRO-3 + CPU Offload | |
| Eval Interval | 256 | |
| | **Model-specific** | |
| | TinyLLaMA-1.1B | LLaMA2-7B |
| #Layers | 22 | 32 |
| #Heads | 4 | 32 |
| Embedding dim | 2048 | 4096 |
| Intermediate dim | 5632 | 11008 |
| Local context length | 2048 | 2048 |

