# OpenReview forum: "UniMem: Towards a Unified View of Long-Context Large Language Models"
_colmweb.org/COLM/2024/Conference — COLM_

### Official Review · Reviewer_pBGi · 2024-05-10

**Rating:** 7
**Confidence:** 4
**Ethics Flag:** 1

**Summary:**

This paper summarized the previous representative works of memory-augmented long LLMs and proposed a unified framework to formulate the methodology of these methods. This paper well resolved the previous issue of incompatible evaluation suites between different method and instead evaluated all methods under the same setting. Additionally, the comprehensive ablation studies are conducted to find the best design choices for each component in the memory module, and the authors used this best hyperparameter groups to train a best model, UniMix.

**Questions To Authors:**

1. How you sample and filter the 0.1B tokens? What are the data resources for that?

2. The number of trainable parameters for different methods vary a lot. Is the same lr 5e-5 fair for the implementation of each method?

3. Why Github is considered as a long-context corpus for evaluation? The more common practice is using ArXiv paper dataset.

**Reasons To Accept:**

1. The research problem is significant to the community. The long-context LLM suffered from the unfair comparison between different methods for quite long.

2. The architecture design lessons and conclusions drawn from the experiments are very useful for other researchers. It helps everyone working on memory-augmented LLMs to have an optimal architecture design.

**Reasons To Reject:**

1. The evaluations are limited to ppl metrics on language modeling dataset. However, the recent research trend in long-context llm community do not regard the lower ppl as the sign for strong long-context understanding capability. I strongly suggested that the authors could consider the e Needle-ina-Haystack tests, the long-document qa experiments, and the long-context in-context learning experiments.

2. The baselines are only trained for 0.1B tokens, which is a small data size for long-context adaptation. Even if the recent consensus on the optimal long-context extension data size has decreased from 100B (Wang et al., 2023) to 5B (Fu et al., 2024), the 0.1B is too small to train the model to understand the long-context input well.

[Wang et al., 2023] Augmenting Language Models with Long-Term Memory
[Fu et al. 2024] Data Engineering for Scaling Language Models to 128K Context

---

> ### Author Rebuttal · Authors · 2024-05-31
>
> We sincerely thank you for your time and efforts. Regarding the weaknesses you mentioned, we are eager to engage in a deeper discussion with you.
>
> ## 1. About adding more downstream evaluation tasks
> Based on your valuable suggestions, we have supplemented downstream experiments on LongBench and "Needles in the Haystack". The results are presented in the table below.
>
> [Table 1: Comparative Performance on Longbench and Table 2: Performance on Needles in the Haystack across different text lengths.](https://anonymous.4open.science/r/Downstream_tasks_results/README.md)
>
> Table 1 shows that the performance ranking aligns with the PPL results: **UniMix > Transformer-XL > Longformer > MemTrans > RMT**.  In Table 2, UniMix is the sole method to identify the "needles" in texts exceeding 4k in length.
>
> ## 2. About the data size of training tokens
> We acknowledge that 0.1B tokens is a small data size. Our decision to use 0.1B tokens in the current experiments was primarily based on the following considerations:
>
> ### (1) Limited Computational Resources
> Our current equipment cannot support multiple model training and tuning on large-scale datasets. The 0.1B tokens data size was a compromise considering training time and cost.
> ### (2) Reference to Existing Work
> We noted that some representative works in long-context modeling, such as Longformer [1], also conducted fine-tuning experiments on datasets of a similar scale.
>
> [1] Longformer: The Long-Document Transformer.
>
> ## 3. About Questions
>
> ### (1) Data size selection
> We selected and filtered 0.1 billion tokens from two primary datasets, the PG-19, and the GitHub datasets, and we focused on instances longer than 20k characters(as illustrated in Section 4.1).
>
> ### (2) Number of trainable parameters and learning rate
> The number of parameters is consistent across different methods, except that RMT introduces a small number of additional parameters. For the learning rate, we found that learning rate does not affect performance a lot.
>
> ### (3) Dataset choice
> GitHub has been used by significant related works like the Memorizing Transformer[2][3]. We plan to include more datasets, such as Arxiv, in the future to enhance the robustness and applicability of our research findings.
>
> [2] Wu et al., Memorizing Transformers.
>
> [3] Tworkowski et al., Focused Transformer: Contrastive Training for Context Scaling.
>
>
> Thank you once again. We look forward to discussing this with you. Please let us know if you have any other questions!

---

> > ### Comment · Reviewer_pBGi · 2024-06-05
> >
> > Thanks for the clarification. The limited computational resource is a large problem for academia research. I hope that you can find more GPUs to consolidate your work and claims via scaling up the training scale. I will keep the rating unchanged.

---

### Official Review · Reviewer_Hfb4 · 2024-05-11

**Rating:** 6
**Confidence:** 3
**Ethics Flag:** 1

**Summary:**

This paper focused on extending the context length of large language models. It firstly reviewed several common existing context modeling approaches in Transformer-based LLMs, including Memory Management, Memory Writing, Memory Reading, and Memory Injection. Then it proposed an unified framework that reformulates existing long-context methods from the view of Memory augmentation of LLMs. It also proposed UniMix, an innovative approach that integrates the strengths of these algorithms. Experimental results show that UniMix achieves superior performance in handling long contexts with significantly lower perplexity than baselines.

**Questions To Authors:**

See above.

**Reasons To Accept:**

1. The unified framework for existing long-context methods is elegant in math.
2. The proposed approach has some advantage on metric of perplexity.

**Reasons To Reject:**

1. The baselines can be supplemented. There are quite a lot of recent works on extending context of LLMs [1][2][3], I would suggest this paper should compare with them.

[1] Pal et al, Giraffe: Adventures in Expanding Context Lengths in LLMs, https://arxiv.org/pdf/2308.10882

[2] Peng et al, YaRN: Efficient Context Window Extension of Large Language Models, https://arxiv.org/pdf/2309.00071

[3] Tworkowski et al, Focused Transformer: Contrastive Training for Context Scaling, https://arxiv.org/pdf/2307.03170

2. The evaluation method can be enhanced. The paper only considered evaluating perplexity of differenct models, however, it ignored some popular downstream tasks tailored for long-context LLMs, such as Needles in a Haystack [4], Document-based Q&A task [5]. I would suggest the author can evaluate their approach in those benchmarks.

[4] Kuratov et al, In Search of Needles in a 11M Haystack: Recurrent Memory Finds What LLMs Miss, https://arxiv.org/pdf/2402.10790

[5] Ma et al, MEGALODON: Efficient LLM Pretraining and Inference with Unlimited Context Length, https://arxiv.org/pdf/2404.08801

---

> ### Author Rebuttal · Authors · 2024-05-31
>
> Thank you very much for your comments! Regarding your valuable suggestions, we here commit to the following:
>
> ## 1. About the supplemented baseline
>
> We agree that Giraffe, YaRN, and Focused Transformer represent the latest advancements in this field, and including them in the baseline comparisons is essential for comprehensively evaluating the performance of UniMix. We will now discuss each of these three models individually:
> - Giraffe [1] and YaRN [2]: Giraffe and YaRN offer insights into positional interpolation and linear scaling methods, which can provide valuable comparisons for UniMix. In future work, We will include a detailed analysis of these position interpolation methodologies and their relationship with the UniMem framework.
> - Focused Transformer [3]: The model can access an additional context of (key, value) pairs through the k-nearest neighbors (kNN) algorithm. Focused Transformer's approach aligns with the Memory Reading and Memory Injection dimensions of the UniMem framework, as it utilizes similarity-based retrieval and augments specific layers with additional context, as shown in Table 1.
> Due to computation resource limitations, we plan to supplement these three models as new baselines in our future work. These works will further refine the UniMem framework, guiding our ongoing efforts to enhance long-context processing capabilities.
>
> [1] Pal et al., Giraffe: Adventures in Expanding Context Lengths in LLMs.
>
> [2] Peng et al., YaRN: Efficient Context Window Extension of Large Language Models.
>
> [3] Tworkowski et al., Focused Transformer: Contrastive Training for Context Scaling.
>
> ## 2. About adding more downstream evaluation tasks
> Based on your valuable suggestions, we have supplemented downstream experiments on LongBench (including a long document QA task) and "Needles in the Haystack". The results are presented in the tables below.
>
> [Table 1: Comparative Performance on Longbench and Table 2: Performance on Needles in the Haystack across different text lengths.](https://anonymous.4open.science/r/Downstream_tasks_results/README.md)
>
> Table 1 shows that the performance ranking aligns with the PPL results: **UniMix > Transformer-XL > Longformer > MemTrans > RMT**.  In Table 2, UniMix is the sole method to identify the "needles" in texts exceeding 4k in length.
>
>
> Thank you once again. Please don't hesitate to reach out if you have any further questions!

---

> > ### Comment · Reviewer_Hfb4 · 2024-06-05
> > **Thanks for rebuttal.**
> >
> > Thanks. I will keep my rating unchanged.

---

### Official Review · Reviewer_HrNV · 2024-05-12

**Rating:** 6
**Confidence:** 4
**Ethics Flag:** 1

**Summary:**

**Summary.** In this paper, the author aim to make a systematic review for the existing techniques on long-context LM. The paper proposes a new framework referred as UniMem, in which four dimensions are introduced for analysis: memory management, memory reading, memory writing, memory injection. The paper also presents a UniMix where the existing memory reading and writing techniques are combined to handle the long-context. Finally the paper conducts empirical study based on several long-context language modeling datasets.

**Reasons To Accept:**

**Strength.** Long-context language modeling is a popular and critical research topic today. Given the quick emergence of new models and algorithms, systematic discussions in this field will be beneficial to the technical community.

**Reasons To Reject:**

**Weakness.** However, there are concerns in the following perspectives regarding the paper's quality.

- The value of the proposed framework is unclear. Although the existing techniques are discussed based on the four dimensions introduced in this paper, very few insights can be drawn from the discussions.

-  The technical novelty of this paper is limited. This paper seems like a research-oriented paper. However, the proposed method is merely a simple combination of the existing approaches.

- The experimental study needs to be improved. Despite that the experiment presents preliminary results on language modeling, none of other important evaluation methods and benchmarks are included, like NIHS, topic retrieval, longbench, et. al.

---

> ### Author Rebuttal · Authors · 2024-05-31
>
> We sincerely thank you for your time and efforts.
>
> ## 1. About the value and novelty of UniMem
> The variety of extensive long-context methods can be challenging for researchers and industry users of this important area. UniMem provides them with a unified memory perspective to systematically understand these approaches. Thus, we think UniMem's value lies in the three aspects:
> ### (1) Better understanding
> UniMem categorizes diverse long-context approaches into four clear memory dimensions. For instance, StreamingLLM and Memorizing Transformer were initially seen as entirely different methods, but from the UniMem perspective, their main difference lies in their use of different memory reading methods (position reading vs. similarity reading). UniMem's four memory aspects give a broad, clear adjustment range for long-context modeling. For instance, from the perspective of Memory Management, researchers can explore various overflow handling strategies, such as Least Recently Used (LRU), to maximize memory capacity utilization. Moreover, Memory Writing can be further optimized by experimenting with different compression and storage formats to achieve various levels of granularity and effectiveness in storage and retrieval.
> ### (2) Insightful findings
> For instance, we find varying the memory injection layer notably affects performance. Each language model has a distinct best injection layer, regardless of the downstream dataset.
> ### (3) Fair evaluation
> UniMem categorizes long-context techniques into four memory dimensions, enabling granular control through tunable hyperparameters (Table 2). This standardization ensures equitable assessments, enhancing the reliability of comparative analyses.
>
> In summary, UniMem is more of an analytical paper rather than solely proposing a stronger combined long-context method like UniMix.
>
> ## 2. About adding more downstream evaluation tasks
> Based on your valuable suggestions, we have supplemented downstream experiments on LongBench. The results are presented in the table below.
>
> [Table 1: Comparative Performance on Longbench](https://anonymous.4open.science/r/Downstream_tasks_results/README.md)
>
> The results show that the performance ranking aligns with the PPL results: **UniMix > Transformer-XL > Longformer > MemTrans > RMT**.
>
> Thank you once again. We look forward to discussing this with you. Please let us know if you have any other questions!

---

> > ### Comment · Reviewer_HrNV · 2024-06-04
> >
> > Thanks for the clarifications! I think the authors do make a hard effort in justifying the value of this paper, though the additional result on Longbench seems too low to present a fully persuasive conclusion.  I appreciate the novel perspective of this paper and hope the authors continue advancing their research. I will give a positive assessment to encourage this kind of work.

---

> ### Author Response · Authors · 2024-06-03
>
> Dear Reviewer, I hope you're doing well. Following up on our recent exchange regarding this paper, I wanted to check if there are any further concerns or feedback from your side. Your insights are invaluable to us, and we're keen to address any remaining issues.

---

### Official Review · Reviewer_95Yk · 2024-05-13

**Rating:** 7
**Confidence:** 4
**Ethics Flag:** 1

**Summary:**

This paper presents a unified framework, UniMem, to reformulate most existing long-context modeling works from a perspective of memory augmentation for LLMs. The paper briefly reviews 4 representative efforts from four memory related dimensions: memory management, writing, reading and injection, in each of which the authors select one model to explain. After analysis regarding the 4 dimensions, the authors develop a new method, UniMix, to benefit from the best features/advantages from the discussed four dimensions. Experiments on two different scales of models, tinyLLaMA-1.1b and LLaMA2-7, to discuss the effect of different long-context solutions as well as position interpolation.

**Questions To Authors:**

please see the comments in the previous section.

**Reasons To Accept:**

1, To consider the long-context modeling works from a unified perspective is new and seems to be inspiring for the community.
2, a new method is proposed based on the analysis using the proposed framework UniMem, which seems to be effective.

**Reasons To Reject:**

1, The existing works covered in the proposed analyzing framework are still limited. For example, many recent works addressing modified position interpolations  are not mentioned or discussed.
2, Using perplexity to evaluate long-context modeling ability solely may not be a choice. The authors should consider using different down streaming tasks, even using \textit{needle in a haystack style} evaluation, which could be a better choice for illustration.
3, Different long-context modeling methods may need different amounts of fine-tuning data, it would be more convincing to plot the trends of using different amounts of fine-tuning data. And it would be better give more about the statistics of the data (there have been several data engineering works to address the importance of fine-tuning data to improve the long-context modeling abilities.
4, The section 4.4 may not be a fair comparison.

---

> ### Author Rebuttal · Authors · 2024-05-31
>
> Thank you very much for your comments! Regarding your valuable suggestions, we here commit to the following:
>
> ## 1. About discussing more recent work about position interpolation
> Following your suggestion, we will add a "Related Work" section to discuss long-context methods parallel to UniMem, especially Position Interpolation (PI). UniMem and PI use different strategies for longer contexts: UniMem optimizes attention mechanism complexity via memory mechanisms, while PI adjusts the Transformer's position encoding for longer contexts than those used in pre-training.
>
> ## 2. About adding more downstream evaluation tasks
> Based on your suggestions, we have supplemented downstream experiments on LongBench. The results are presented in the table below.
>
> [Table 1: Comparative Performance on Longbench](https://anonymous.4open.science/r/Downstream_tasks_results/README.md)
>
> Due to time limitations, our experiments were performed on tinyLlama, setting the context length to 512 and the memory size of Unimix to 10k tokens, while other hyperparameters were reduced proportionally compared to those specified in the paper.
>
> The above results show that the performance ranking of downstream tasks aligns with the PPL evaluation in our paper: **UniMix > Transformer-XL > Longformer > MemTrans > RMT**.
>
> We also conducted experiments using the "Needles in the Haystack" task, which comprises text lengths ranging from 4k to 32k, as depicted in the table below. UniMix is the sole method capable of identifying the "needles" in texts exceeding 4k in length.
>
> [Table 2: Performance on Needles in the Haystack across different text lengths](https://anonymous.4open.science/r/Downstream_tasks_results/README.md)
>
> ## 3. About supplementary analysis of the impact of different training data scales.
>
> Based on your suggestion, we explored the impact of varying fine-tuning data volumes on long-context modeling methods. We plotted the Perplexity (PPL) for TinyLLAMA and LLAMA2-7B using different scales of training data, as shown in the figure below.
>
> [PPL trend of TinyLLaMA and  LLaMA2-7B on the GitHub Dataset](https://anonymous.4open.science/r/exp_mem-2EF0/README.md)
>
> We will include the above experimental results in the paper.
>
> ## 4. About the concern of Section 4.4
> Could you provide more detailed suggestions or guidelines? Specifically, how should we design our experiments to address your concerns?
>
> Thank you once again. Please don't hesitate to reach out if you have any further questions!

---

> > ### Comment · Reviewer_95Yk · 2024-06-07
> > **Thanks for the response**
> >
> > Thanks for the response which addresses most of my questions.
> > I will raise my scores.

---

### Decision · Program_Chairs · 2024-07-10

**Decision:**

Accept

**Comment:**

The paper introduces a unified framework for memory augmentation of LLMs. Long-term memory is an important topic for LLM research. The overall rating turns out to be positive. It would be good to include the submission in the conference proceedings.